# Towards Understanding Non-Infectious Growth-Rate Retardation in Growing Pigs

**DOI:** 10.3390/proteomes7030031

**Published:** 2019-09-11

**Authors:** Ana M. Gutiérrez, Juan Sotillo, Sarah Schlosser, Karin Hummel, Ingrid Miller

**Affiliations:** 1Department of Animal Medicine and Surgery, BioVetMed research group, University of Murcia, 30100 Murcia, Spain; jsotillo@um.es; 2VetCore Facility for Research, University of Veterinary Medicine of Vienna, A-1210 Vienna, Austria; Sarah.Schlosser@vetmeduni.ac.at (S.S.); Karin.Hummel@vetmeduni.ac.at (K.H.); 3Institute of Medical Biochemistry, University of Veterinary Medicine of Vienna, A-1210 Vienna, Austria; Ingrid.Miller@vetmeduni.ac.at

**Keywords:** growth-rate retardation, gel-based proteomics, pig, saliva, biomarker detection

## Abstract

For growth-rate retardation in commercial growing pigs suffering from non-infectious diseases, no biomarker is available for early detection and prevention of the condition or for the diagnosis of affected animals. The point in question is that the underlying pathological pathway of the condition is still unknown and multiple nutritional or management issues could be the cause of the disease. Common health status markers such as acute phase proteins, adenosine deaminase activity or total antioxidant capacity did not show any alteration in the saliva of animals with growth-rate retardation, so other pathways should be affected. The present study investigates saliva samples from animals with the same commercial crossbreed, sex and age, comparing control pigs and pigs with growth-rate retardation. A proteomics approach based on two-dimensional gel electrophoresis including mass spectrometry together with validation experiments was applied for the search of proteins that could help understand disease mechanisms and be used for early disease detection. Two proteins were detected as possible markers of growth-rate retardation, specifically S100A12 and carbonic anhydrase VI. A decrease in innate immune response was confirmed in pigs with growth-rate retardation, however further studies should be necessary to understand the role of the different CA VI proteoforms observed.

## 1. Introduction

Growth-rate retardation in porcine production is a problem that causes serious economic losses to stockbreeders. The weight gain in affected animals does not follow the expected time course. Therefore, they stay smaller and with lower weight than their fellow beings. The parameter average daily gain (ADG) is used to numerically assess animal growth and it gives the weight gained per day, expressed in grams/day. The ADG may be low, moderate or ideal, and there are different referential models that correspond to each one of these situations. In optimal conditions, the expected ADG is from 614 to 682 g/day during the fattening phase [1].

In piglets, growth-rate retardation is due to a delay in their growth during the prenatal phase, due to a restriction in the intrauterine growth [2]. In fattening animals, there are several causes that could act individually or in combination, such as non-infectious factors, that include genetics or immune status, and infectious factors, including pathogenic parasites, bacteria and viruses [3]. There are methods to confirm the etiologic diagnosis for genetic and infectious causes, but in some animals with growth-rate retardation it is not an easy task to recognize the reasons for the process, due to their multifactorial character.

The number of pigs per pen, the type of feeder and the mix of animals of different origin or sex have been considered as the main parameters affecting the daily feed intake and the feed conversion ratio of pigs at growing phase [4]. Low protein supply in growing pigs has been reported to have a retardation effect due to lower muscle growth [5]. Moreover, it has been indicated that improving farm facilities and modifying management practice could reduce mortality and increase growth performance of grower-finishing pigs [6]. A detrimental effect of low temperature has been evidenced on performance in pigs with restricted diets [7].

The combination between infection with Porcine Circovirus type 2 (PCV2) and environmental stress has been reported to lead to decreased ADG [8]. However, other studies have not obtained a connection between low growth and PCV2 virus in grower-finishing pigs but an influence of diarrhea [9].

The present study is focused on the identification of proteins altered in pigs under growth-rate retardation using a gel-based proteomics approach in saliva samples, as they are supposed to show up affected pathways in this disease/syndrome. 

## 2. Material and Methods

### 2.1. Characterization of Animals

20 commercial males ((Landrace x Large White) x Duroc) growing pigs were selected from a previous study [10]. Two groups of animals were randomly selected based on clinical signs and availability of saliva samples: a group of 10 healthy animals without any clinical sign of disease (control pigs), and a group of 10 animals with exclusively growth-rate retardation (GRR) symptoms (GRR pigs).

Saliva samples were collected individually, as reported in the mentioned study, during clinical veterinary inspection of the animals by allowing them to chew a sponge. Afterwards the sponges were centrifuged in a specifically designed tube (Salivette tubes, Salimetrics, USA). Salivary supernatants were stored at −80 °C until analysis.

To exclude any subclinical infection against Porcine Circovirus type 2 (PCV2), which is a common cofactor in the development of several syndromes in the porcine production system, PCR analysis was performed on the saliva samples. 

To obtain an overview of the possible mechanism that could be involved in the growth-rate retardation process, the acute phase reaction, the inflammatory and antioxidant status and the level of stress were determined in all pigs.

### 2.2. PCV2 Detection

The primers used for PCV2 detection in porcine saliva samples were reported previously [11], specifically Forward primer: 5′-CGGATATTGTATTCCTGGTCGTA-3′ and Reverse primer: 5′-CCTGTCCTAGATTCCACTATTGATT-3′. The extraction of DNA from saliva samples was performed automatically using a commercial extraction kit in an automated DNA extraction instrument (Maxwell^®^ 16 Blood DNA Purification Kit&Maxwell^®^ RSC Instrument for 16 samples, Promega Corporation, Madison, WI, USA, respectively). The extracted DNA samples were subjected to RT-PCR using an intercalating dye-based qPCR master mix (TB Green Premix Ex Taq (Tli RNase H Plus), Takara Bio Inc., Saint-Germain-en-Laye, France) and a real-time PCR system (Lightcycler 2.0. Roche Diagnostics, Indianapolis, IN, USA).

### 2.3. Salivary Marker Measurements

C-reactive protein (CRP) and haptoglobin (Hp) concentrations were determined in the saliva samples, for identification of acute and chronic acute phase reaction, respectively, using time-resolved immunofluorimetric assays described before [12,13].

For the inflammatory and antioxidant status measurement, adenosine deaminase (ADA) activity and total antioxidant status (TAC) were determined, respectively, using enzymatic assays as reported in previous studies [10,14].

The stress status in the selected animals was evaluated by measuring the alpha-amylase concentrations in the saliva samples with a commercial enzymatic assay for the human homologue, which had been previously optimized for pigs [15].

The possible differences in the levels of the markers studied between control pigs and GRR pigs were statistically evaluated using a Mann-Whitney *t*-test, since data did not meet the normal distribution criteria. Any possible correlation between the studied analytes were detected by Spearman correlation test. The level of significance was set at *p* < 0.05 with the statistical program GraphPad Prism 6 (Graph Pad Software Inc., La Jolla, CA, USA).

### 2.4. Two-Dimensional Gel Electrophoresis (2DE)

To perform the protein profile comparison between control pigs and pigs with GRR, 2DE was used as previously described [16]. Briefly, 30 μg of saliva proteins in rehydration buffer were rehydrated into immobilized pH gradient (IPG) strips with 11 cm long nonlinear gradients pH 3–11 (GE Healthcare Life Sciences, Munich, Germany) and subjected to isoelectric focusing in a Multiphor II electrophoresis chamber (GE Healthcare Life Sciences, Munich, Germany). For the second dimension, the IPG strips were equilibrated with a 2% DTT solution, followed with a 2.5% of iodoacetamide solution, and subjected to SDS-PAGE on homemade 10–15% polyacrylamide gradient gels of 140 × 140 × 1.5 mm in a vertical chamber (SE600 Chroma, Hoefer, INC., Holliston, MA, USA). Protein patterns were silver-stained according to general protocols [17] and scanned in an image scanner (ImageScanner II, GE Healthcare Life Sciences. Uppsala, Sweden). Images were evaluated for spot detection and matching using specific software (ImageMaster 2D Platinum 7.0, GE Healthcare Life Sciences, Uppsala, Sweden). The relative percentage of spot volumes from each group of animals was statistically compared using a *t*-test with the software mentioned above.

Selected spots with differential abundance between groups were subjected to mass spectrometric (MS) identification.

### 2.5. Sodiumdodecyl Sulphate Polyacrylamide Gel Electrophoresis (SDS-PAGE)

One-dimensional SDS-PAGE was done in parallel to the 2DE approach in order to support those data and to see whether additional differences between the sample groups could be picked up. After protein determination [18] 2.5 μg of total protein of each saliva sample were electrophoretically separated by SDS-PAGE as described for the second dimension of 2DE. Following silver staining and scanning, gel images were evaluated with a specific software (ImageQuant TL v2005, Amersham Biosciences Europe GmbH, Freiburg, Germany), to obtain the relative percentage in volume of the different bands in the gel. To evaluate any possible band volume differences between groups of animals, a *t*-test was used with the statistical software detailed above. Corresponding bands differing by intensity were subjected to MS identification.

### 2.6. MS Identification of SDS-PAGE Bands and 2DE Spots

After washing and destaining, bands or spots were reduced with dithiothreitol and alkylated with iodoacetamide [19]. In-gel digestion was performed with trypsin (Trypsin Gold, Mass Spectrometry Grade, Promega, Madison, WI) with a final trypsin concentration of 20 ng/µL in 50 mM aqueous ammonium bicarbonate and 5 mM CaCl_2_ [20]. Afterwards, peptides were extracted with four changes of 30 µL of 5% trifluoroacetic acid (TFA) in 50% aqueous acetonitrile (ACN) supported by ultrasonication for 10 min per change and dried down in a vacuum concentrator (Eppendorf, Hamburg, Germany).

### 2.7. Protein Identification by MALDI-TOF/TOF Mass Spectrometry

Dried peptides were concentrated and de-salted using Zip-Tips C18 (microbed) (Millipore, Billerica, MA) according to the manufacturer’s instructions.

After elution from the Zip-Tip 0.5 µL of the de-salted peptides were spotted with α-cyano-4-hydroxycinnamic acid onto a ground steel MALDI target plate (Bruker Daltonics, Bremen, Germany). MALDI-TOF/TOF mass spectrometry (Ultraflex II, Bruker Daltonics, Bremen, Germany) was used for spectra acquisition in MS and MS/MS modes. Spectra processing and peak annotation were carried out using FlexAnalysis 3.0 and Biotools 3.2 (both Bruker Daltonics, Bremen, Germany).

Processed spectra were searched via an in-house Mascot server version 2.4.1 (Matrix Science, Boston, MA) and the software ProteinScape 2.1 (Bruker Daltonics, Bremen, Germany) in the UniProt database of *sus scrofa* using the following search parameters: global modification carbamidomethylation on cysteine; variable modifications oxidation on methionine; deamidation on asparagine and glutamine as well as formation of pyroglutamic acid; enzyme specificity trypsin; charge state z = 1; MS tolerance 100 ppm; MS/MS tolerance 1 Da; two missed cleavages allowed; significance threshold *p* < 0.05.

### 2.8. Protein Identification by Q Exactive HF Orbitrap Mass Spectrometry

Dried peptides were resuspended in 0.1% TFA for the LC-MS/MS analysis and separated on a nano-HPLC Ultimate 3000 RSLC system (Dionex). Sample pre-concentration and desalting was accomplished with a 5 mm Acclaim PepMap μ-Precolumn (Dionex). For sample loading and desalting 2% ACN in ultra-pure H_2_O with 0.05% TFA was used as a mobile phase with a flow rate of 5 µL/min. Separation of peptides was performed on a 25 cm Acclaim PepMap C18 column with a flow rate of 300 nL/min. The gradient started with 4% B (80% ACN with 0.08% formic acid) and increased to 9% B in 7 min, to 31% in 30 min and to 44% in additional 5 min. It was followed by a washing step with 95% B, Mobile Phase A consisted of ultra-pure H_2_O with 0.1% formic acid.

For MS analysis, the LC was directly coupled to a high-resolution Q Exactive HF Orbitrap mass spectrometer. MS full scans were performed in the ultrahigh-field Orbitrap mass analyzer in ranges *m*/*z* 350−2000 with a resolution of 60,000, the maximum injection time (MIT) was 50 ms and the automatic gain control (AGC) was set to 3e6. The top 10 intense ions were subjected to Orbitrap for further fragmentation via high energy collision dissociation (HCD) activation over a mass range between *m*/*z* 200 and 2000 at a resolution of 15,000 with the intensity threshold at 4e3. Ions with charge state +1, +7, +8 and >+8 were excluded. Normalized collision energy (NCE) was set at 28. For each scan. the AGC was set at 5e4 and the MIT was 50 ms. Dynamic exclusion of precursor ion masses over a time window of 30 s was used to suppress repeated peak fragmentation.

Database search was performed using an in-house Mascot server (version 2.4.1., Matrix Science, Boston, MA) with following settings:

The protein database consisted of amino acid sequences downloaded from UniProt for taxonomy “*Sus scrofa*” (TaxID 9823) as well as a common contaminant database (http://www.thegpm.org/crap/, identified contaminant proteins were removed from results file). Search settings were as described above adapted to the QExactive MS system: global modification carbamidomethylation on cysteine; variable modifications oxidation on methionine; deamidation on asparagine and glutamine as well as formation of pyroglutamic acid; enzyme specificity trypsin; charge state z = 2+, 3+, 4+; MS tolerance 10 ppm; MS/MS tolerance 0.05 Da; two missed cleavages allowed; significance threshold *p* < 0.05.

Finally, taking into account the identified proteins, function information was annotated from UniProt database [21]. 

### 2.9. Protein Quantification by Western Blot

One of the proteins found regulated in SDS-PAGE, S100A12, was subjected to validation analysis. Since the alignment between the human and porcine protein sequences gave an identity of 70.6% according to UniProt database (identification numbers P80511 and P80310, respectively) a commercial human antibody was selected to perform the validation study. Saliva samples of the 10 control pigs and the 10 GRR animals and an additional group of 7 pigs with GRR from the same farm were used for validation. A total protein content of 2.5 μg was applied for SDS-PAGE. Control sample number 6 was randomly selected and used as internal control sample in all gels.

Immunodetection was performed after SDS-PAGE on small-size home-made 12% polyacrylamide mini gels using a vertical electrophoresis chamber (Mini-PROTEAN Tetra Vertical Electrophoresis Cell, Bio-Rad, Hercules, CA, USA) and protein transfer to PVDF membranes using the Trans-Blot Turbo Transfer System (Bio-Rad, Hercules, CA, USA). The primary antibody applied was goat anti-human S100A12 (Thermo Scientific, Rockford, IL, USA) followed by horseradish peroxidase conjugated rabbit anti-goat IgG (Sigma Aldrich, St. Louis, MO, USA). Positive signal was detected by enhanced chemiluminescence (ECL 2 Western Blotting Substrate, Thermo Scientific, Rockford, IL, USA) in an imager LAS 600 (GE Healthcare, Uppsala, Sweden). The comparison between the signals of the different samples was performed using an image analysis software (ImageQuant TL v2005, Amersham Biosciences Europe GmbH, Freiburg, Germany). For comparison of results from different gels, the signal intensity of the internal control sample was used for data normalization.

## 3. Results

### 3.1. PCV2 Detection

All saliva samples from control pigs and GRR pigs appeared negative to PCV2 after RT-PCR analysis (Appendix A) since no positive signal at the expected 149 bp was observed.

### 3.2. Salivary Measurements

The concentrations of acute phase proteins, specifically CRP and Hp, as well as the activity levels of ADA and TAC in the two groups of study were similar, with no statistical differences (Table 1).

The levels of alpha amylase in the group of GRR pigs were slightly higher than those observed in control animals, median values of 179.7 U/mL vs. 44.93 U/mL respectively, but without statistical significance (*p* > 0.05).

A positive correlation was observed between the acute phase proteins Hp and CRP and between the CRP and TAC and amylase activity. Moreover, a negative significant correlation was observed between ADA activity and CRP and TAC (Table 2).

### 3.3. DE Analysis

A total of 11 spots appeared to be differentially regulated in GRR in comparison to control pigs (Figure 1, Table 3).

Six out of the 11 listed spots were up-regulated in GRR pigs and were identified as immunoglobulin single chains or fragments (light chains or IgA constant region) and carbonic anhydrase VI. The other 5 proteins were up-regulated in control pigs and were identified as salivary lipocalin, odorant binding protein, double-headed protease inhibitor submandibular gland-like protein and carbonic anhydrase VI (a spot of lower MW). For three spots (numbers 50, 157 and 156), protein content was too low to obtain a valid identification by MALDI-TOF-TOF, therefore an LC-MS/MS approach was used for protein identification.

Carbonic anhydrase VI was detected in two spot chains of different molecular weight, the larger one up- and the lower one down-regulated in animals with GRR. Identification was done in the main spots of these chains (spot numbers 54, 55, 62), but also the other spots showed a similar trend in regulation, though not all did reach statistical significance (Figure 2).

### 3.4. SDS-PAGE Analysis

After the analysis of all saliva samples by SDS-PAGE, five bands were observed to be differentially regulated between the two groups of study (Figure 3).

Two of them were up-regulated in GRR pigs (band numbers 36 and 46) and the other three were up-regulated in control animals (band numbers 39, 40 and 49). Due to the high sensitivity of the LC-MS/MS system used, the MS identification of the differentially regulated bands (Appendix A) revealed a complex mix of proteins, with up to 30–70 identified proteins (section criteria: at least 2 identified peptides). Exclusion of contaminating proteins (based on the cRAP-database) and rigorous sorting were tried to limit hits to the major proteins contributing to the staining in the respective bands, as those were assumed the most likely responsible for band intensity. Sorting was performed according to exponentially modified protein abundance index (emPAI) (cutoff: < 10% of highest achieved identification).

Based on emPAI filtering (Table 6), three of the five bands showed a considerable contribution of salivary lipocalin (namely salivary lipocalin 1 (SAL 1)), or “Epididymal-specific lipocalin-9 isoform X1”, both identified from the same peptides, one of them additionally Ig light chains (lambda) and Ig-like domain containing proteins. Band number 46, up-regulated in GRR pigs, displayed as a main protein cystatin-C precursor besides some hemoglobin. The down-regulated band with the lowest kDa-value presented as best hits two calcium-binding S100 proteins, namely S100A12 (highest emPAI) followed by S100A8.

Though separating proteins based on only one parameter (protein or subunit size) can give only limited specific information, in combination with extensive emPAI filtering it revealed the most likely regulated proteins in some of the bands, confirming 2DE data.

Based on the regulation data of band 49 where S100A2 was identified with good scores, a validation by immunoblotting was undertaken. It confirmed decrease of band intensity of this protein in pigs with GRR in comparison to control animals (Figure 4) by a factor of 2.2 when the same samples as in the proteomic approach were used. Moreover, the addition of 7 new samples from GRR animals of the same farm confirmed the down-regulation behavior of S100A12 in this pathophysiological condition, at least in the farm studied.

## 4. Discussion

The focus for our study was on obtaining additional information about growth-rate retarded pigs without infectious disease, as too little biological data is available for this pathophysiological condition.

No differences in the level of acute phase proteins nor in the concentrations of ADA or TAC activity were observed between the two groups of animals. Similar results were also previously described in another study that involved a higher number of animals [10]. A possible subclinical infection against PCV2 could be excluded. Moreover, a trend to increased salivary alpha-amylase concentrations in animals with GRR was observed in comparison to the control pigs but without statistical significance. A higher level of stress could be postulated in animals suffering from GRR and could be considered as one of the factors affecting the animals, in accordance to [6].

To search for biological pathways that could be involved in GRR development, proteins changed in abundance were studied in saliva samples by a 2DE gel-based proteomic approach. Among the regulated proteins, Salivary lipocalin (SAL), double-headed protease inhibitor submandibular gland-like protein and Odorant binding protein (OBP) were detected as down-regulated proteins, whereas lambda light chain of immunoglobulins as well as a fragment of IgA were up-regulated in GRR.

SAL is synthetized mainly in the boar submaxillary gland and binds reversibly odorants and the endogenous ligands androstenol and androstenone [22]. The protein has two isoforms that are glycosylated at a single position [23]. SAL is a member of the lipocalin family in which OBP is also included. OBPs are assumed to be directly involved in chemical communication and in the pre-mating recognition process [24]. Since those proteins were observed down-regulated in GRR pigs it could be supposed that sexual communication may be sub-optimal in those animals. Our SDS-PAGE results support the 2DE findings of altered SAL concentrations; this protein was found as a main component of two bands with down-regulated intensities in GRR. Previous own experiments to detect pig salivary lipocalin more specifically with an antibody against the human homologue failed (the antibody was produced against a synthetic peptide with a sequence close to the C-terminal end of the protein; Sigma Aldrich). The reaction of the antibody was not stable over time, though the reason could not be traced back to either the immunoreagent or the sample (in)stability [25].

Carbonic anhydrase VI was another differentially regulated protein of our study, found in different locations of the gel and in altered abundance. It has been reported that CA VI participates in the regulation of salivary pH and protects oral tissue against excess acidity [26]. Two different forms of 41 kDa and 37 kDa have been reported in rat whole saliva corresponding to carbonic anhydrase VI and a partially deglycosylated form of CA VI [27]. We have also obtained two forms of CA VI in porcine saliva with similar molecular weights in the present study. Moreover, a higher level of CA VI in its bigger form appeared in animals with GRR while a lower level of the small molecular weight CA VI form was also observed. Further studies should be necessary to understand the different forms of CA VI and its relation to GRR. It could be postulated that the decrease in the smaller, assumed partially deglycosylated form in pigs with GRR might be related to a higher stress status since it has been reported that stress conditions produce an intracellular form of CA VI in rats [28]. In saliva samples from animals under stress conditions this protein form could be (almost completely) lacking.

Both, 2DE and SDS-PAGE, detected upregulation of immunoglobulin light chains or parts of heavy chains. This could reveal a humoral immune activation in GRR pigs as reported in other porcine diseases [29]. However, markers of innate immune response such as acute phase proteins, adenosine deaminase and S100A12 levels appeared decreased in those pigs with GRR, so further studies should be performed to explain the immune reaction status in pigs with GRR.

Evaluation of band 49 of the SDS-PAGE pattern suggested regulation of two S100 proteins, S100A8 and S100A12. Only for S100A12 a cross-reactive antibody was available, confirming its downregulation in animals with GRR. This was found both when investigating the 10 saliva samples originally included in the 1DE and 2DE sets and when adding 7 samples from other GRR animals of the previous trial [10]. S100A12 has proinflammatory activity and is involved in recruitment of leukocytes, promotion of cytokine and chemokine production, and regulation of leukocyte adhesion and migration (UniProt database). According to these results it could indicate that animals with GRR may be more susceptible to infections with an increase in stress response. Validation of S100A8 data was also tried by western blot in our study. However, the available human S100A8 antibody (Thermo Scientific, Rockford, IL, USA) used did not show any cross-reactivity against the porcine protein. This may be due to differences of the human and porcine sequences (identity of 68.8% between the identification numbers P05109 and C3S7K5, respectively, UniProt database).

Developing targeted proteomic assays (selected reaction monitoring, SRM) for the proteins found regulated in this study could be a way to avoid searching for well-reacting commercial pig-specific or cross-reactive antibodies. SRMs are based on MS detection of unique peptide sequences and their fragmentation pattern (transitions). Candidate peptides may be selected either from the peptides identified in the respective study, or in comparison with dedicated database collections [30], and, once successfully set up, could also in our case help to verify protein candidates or screen larger numbers of samples.

In conclusion, we have detected two possible proteins that could be used as markers of growth-rate retardation in pigs, S100A12 and CA VI, and should be explored in-depth for routine quantification. The behavior of S100A12 in the conditions studied confirmed that the innate immune response is decreased in pigs with GRR. However, on the other hand, concentration of some immunoglobulin chains increased—suggesting some imbalance of the immune system. The role of CA VI in the disease should be clarified in future studies. It may additionally be advisable to set up a specific detection system for the different forms of this protein as well as characterizing them in more details. Moreover, a stress influence was also suggested in animals suffering from GRR that should be taken into account in further studies.

## Figures and Tables

**Figure 1 proteomes-07-00031-f001:**
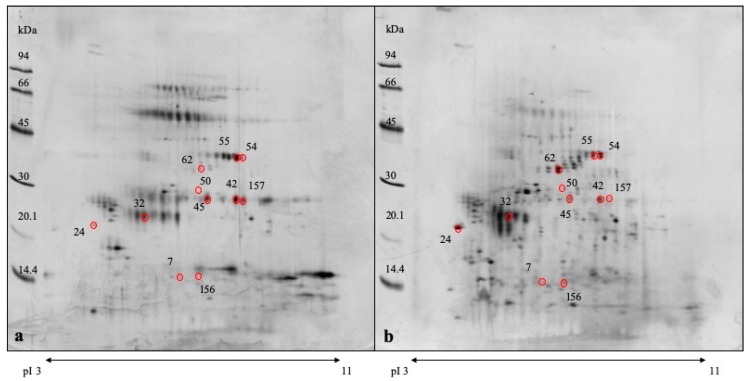
Gel images of porcine saliva samples from a pig with growth-rate retardation (**a**) and a control pig (**b**). Red marked spots show statistically significant changes in abundance between the two health status conditions. For details on spot changes and protein identifications see Table 3, Table 4 and Table 5, respectively.

**Figure 2 proteomes-07-00031-f002:**
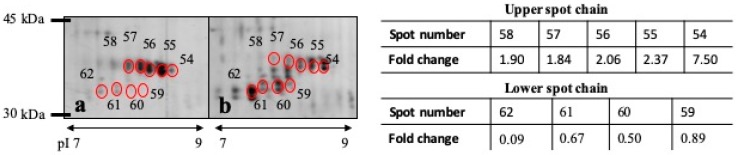
Close-up of the 2-DE region of pH 7–9 and MW 45–30 kDa for a pig with growth-rate retardation (**a**) and a control pig (**b**). Red marked spots (54–62) correspond to carbonic anhydrase VI protein. Fold change of carbonic anhydrase VI spots chains. Fold change: ratio mean value in GRR/mean value in control pigs. Mean value: percentage of spot volume.

**Figure 3 proteomes-07-00031-f003:**
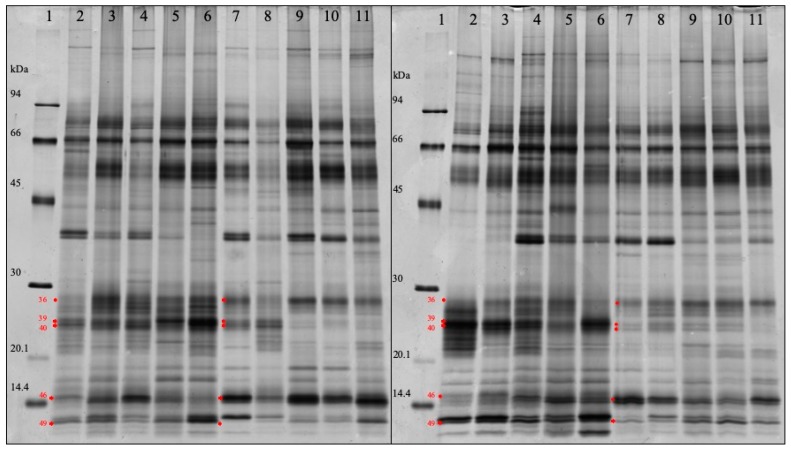
SDS-PAGE image of a group of 10 control pigs (lanes 2–6 of each gel) and a group of 10 pigs with growth-rate retardation (lanes number 7–11 of each gel). Lanes 1 show molecular weight markers in kDa. Red arrows show statistically significant changes in band abundance between the two groups of animals that were subjected to MS analysis for protein identification. For details of band changes and protein identifications see Table 5 and Appendix A, respectively.

**Figure 4 proteomes-07-00031-f004:**
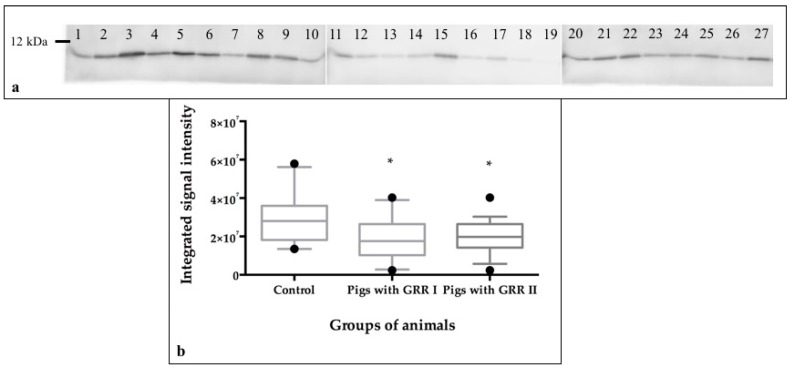
Signal detection after western blot analysis against S100A12 protein in control pigs (lane number 1–10) and pigs with growth-rate retardation (lane numbers 11–27; lane numbers 11–20 represent GRR pigs used in the proteome approach while lane numbers 21–27 show results of an additional group of 7 GRR pigs from the same farm) (**a**). Levels of the integrated signal intensity observed in control pigs and pigs with growth-rate retardation (Pigs with GRR I *n* = 10 and Pigs with GRR II *n* = 17) (**b**). Data were normalized using an internal control sample. The plot shows median (line within box), 25th and 75th percentiles (box), 5th and 95th percentiles (whiskers) and outliers (•). Asterisk represents the statistically significant differences between groups of pigs. * *p* < 0.05. To see the whole WB image and the membrane staining after WB analysis see Appendix A.

**Table 1 proteomes-07-00031-t001:** Concentrations of C-reactive protein (CRP), haptoglobin (Hp), adenosine deaminase (ADA) and total antioxidant status (TAC) in saliva of control pigs (*n* = 10) and in pigs with growth-rate retardation (GRR) (*n* = 10). * Median value (25th–75th percentiles).

Analyte	Control Pigs *	Pigs with GRR *	*p* Value
**CRP (ng/mL)**	3.51 (2.42–12.82)	17.94 (2.42–32.26)	0.2225
**Hp (µg/mL)**	0.45 (0.30–0.77)	0.42 (0.23–0.96)	0.9575
**ADA (U/L)**	269.7 (171.5–344.9)	245.5 (82.53–619.3)	0.9502
**TAC (M trolox equivalents/mL)**	0.66 (0.24–0.90)	0.49 (0.35–1.25)	0.6685

**Table 2 proteomes-07-00031-t002:** Spearman correlation coefficients between all the different markers measured in the saliva samples of control pigs (*n* = 10) and GRR pigs (*n* = 10). * with a level of significance *p* < 0.05.

	CRP	ADA	TAC	Amylase
**Hp**	0.63 *	−0.22	0.14	0.31
	**CRP**	−0.46 *	0.44 *	0.49 *
		**ADA**	−0.69 *	−0.13
			**TAC**	0.09

**Table 3 proteomes-07-00031-t003:** Spots differentially regulated in control pigs in comparison to pigs with growth-rate retardation after Two-Dimensional Gel Electrophoresis (2DE) as seen in Figure 1. Spots sorted by p value. ^a^ protein MS identification appears in Table 4. ^b^ Known function according to UniProt database. ^c^ Fold change: ratio mean value in GRR/mean value in control pigs. Mean value: percentage of spot volume. MW: measured molecular weight in kDa.

Spot Number ^a^	*p* Value	Protein Name	Known Function ^b^	Fold Change ^c^	MW
**42**	0.0018	Ig lambda chain C region	Miscellaneous in the humoral immunity	1.795	27 kDa
**156**	0.0068	Double-headed protease inhibitor submandibular gland-like	Hydrolase, protease activity	0.593	14 kDa
**24**	0.0080	Odorant-bindingprotein	Chemical odorants binding	0.338	24 kDa
**157**	0.0087	Ig lambda chain C region	Miscellaneous in the humoral immunity	2.072	27 kDa
**62**	0.0126	Carbonicanhydrase VI	Carbonate dehydratase activity, zinc ion binding	0.094	33 kDa
**7**	0.0127	Salivarylipocalin	Pheromonebinding	0.468	14 kDa
**45**	0.0132	Ig lambda chain C region	Miscellaneous in the humoral immunity	1.594	27 kDa
**50**	0.0272	IgA heavy chain constant region	Antigen binding in the humoral immunity	1.656	27 kDa
**32**	0.0374	Salivarylipocalin	Pheromonebinding	0.452	25 kDa
**54**	0.0383	Carbonicanhydrase VI	Carbonate dehydratase activity, zinc ion binding	7.509	36 kDa
**55**	0.0492	Carbonicanhydrase VI	Carbonate dehydratase activity, zinc ion binding	2.370	36 kDa

**Table 4 proteomes-07-00031-t004:** Protein identification details of the 2DE spots that appeared differentially regulated between control pigs and pigs with growth-rate retardation (Figure 1). Identifications performed by MALDI TOF-TOF.

Spot ID	Accession	Protein Identified	MW [Da]	pI	Mascot Score	Nr of Peptides	Sequ. Cov %	*m*/*z*	Range	Peptide Score	Peptide Sequence
**7**	P81608	**Salivary lipocalin**	21.6	5.0	184.7	3	20.4	1291.5897	132–141	68.7	K.TFQLMEFYGR.K
								1497.7441	153–164	64.8	K.FVEICQQYGIIK.E
								1938.9012	94–110	51.2	K.VGDGVYTVAYYGENKFR.L
**24**	P81245	**Odorant-binding protein**	17.7	4	607.8	6	54.1	1017.5580	51–58	52.4	K.VYLNFFSK.E
								1408.7844	29–40	101.6	K.IGENAPFQVFMR.S
								1498.8065	16–28	124.7	K.WITSYIGSSDLEK.I
								1711.8749	1–15	81.4	-.QEPQPEQDPFELSGK.W
								2023.0754	121–137	143.1	K.GTDIEDQDLEKFKEVTR.E
								2297.2219	138–157	104.6	R.ENGIPEENIVNIIERDDCPA.-
**32**	P81608	**Salivary lipocalin**	21.6	5.0	469.4	5	26.2	898.5025	59–65	31.8	R.VFVEHIR.V
								1291.6775	132–141	82.6	K.TFQLMEFYGR.K
								1497.7926	153–164	96.4	K.FVEICQQYGIIK.E
								1938.0475	94–110	115	K.VGDGVYTVAYYGENKFR.L
								1982.1818	149–164	143.6	K.LKDKFVEICQQYGIIK.E
**42**	P01846	**Ig lambda chain C region**	11.0	7.5	162.3	2	31.4	1632.8567	65–79	108.6	K.YAASSYLALSASDWK.S
								1967.9933	80–97	53.7	K.SSSGFTCQVTHEGTIVEK.T
**45**	P01846	**Ig lambda chain C region**	11.0	7.5	127.5	2	31.4	1632.8024	65–79	58.9	K.YAASSYLALSASDWK.S
								1966.9425	80–97	68.6	K.SSSGFTCQVTHEGTIVEK.T
**54**	B7X727	**Carbonic anhydrase VI**	36.3	6.2	230.1	3	8.8	1047.5925	56–64	21.2	K.SVQYNPALR.A
								2267.1851	286–303	153.1	R.SELHFYLNNIDNNLEYLR.R
								2423.3125	286–304	55.7	R.SELHFYLNNIDNNLEYLRR.V
**55**	B7X727	**Carbonic anhydrase VI**	36.3	6.2	293.8	4	15.1	1047.6041	56–64	36	K.SVQYNPALR.A
								1074.5952	258–265	30.3	K.TIHNDYRR.T
								1574.9181	133–145	85.6	R.YVTEVHVVHYNSK.Y
								2267.2771	286–303	141.9	R.SELHFYLNNIDNNLEYLR.R
**62**	B7X727	**Carbonic anhydrase VI**	36.3	6.2	470.6	6	24.9	1047.5475	56–64	30.7	K.SVQYNPALR.A
								1074.5327	258–265	35.3	K.TIHNDYRR.T
								1167.4833	38–46	31.4	R.EYPDCDGRR.Q
								1574.7672	133–145	87.7	R.YVTEVHVVHYNSK.Y
								1965.8298	170–185	109.3	K.DYAENTYYSDFISHLK.N
								2699.2092	109–132	176.2	K.QMHFHWGGAFSEISGSEHTIDGIR.Y

**Table 5 proteomes-07-00031-t005:** Protein identification details of the 2DE spots that appeared differentially regulated between control pigs and pigs with growth-rate retardation (Figure 1). Identifications performed by LC-MS/MS. #: number.

Spot ID	Accession	Description	Coverage [%]	# Peptides	# PSMs	# Unique Peptides	# AAs	MW [kDa]	calc. pI	Score Sequest HT:	# Peptides (by Search Engine): Sequest HT
**50**	K7ZRK0	**IgA heavy chain constant region (Fragment)**	15	4	17	4	341	36.6	6.13	28.67	4
**156**	A0A287ATY8	**62.7% ID with double-headed protease inhibitor. submandibular gland-like (Odobenus rosmarusdivergens)**	12	2	2	2	134	14.7	7.02	0	2
**157**	P01846	**Iglambdachain C region**	30	5	48	4	260	27.5	7.65	119.64	5

**Table 6 proteomes-07-00031-t006:** Bands differentially regulated in control pigs in comparison to pigs with growth-rate retardation after SDS-PAGE (Figure 3). Band number sorted by fold change. ^a^ Fold change: ratio mean value in GRR/mean value in control pigs. Mean value: percentage of spot volume. MW: measured molecular weight in kDa. For more details of protein identifications see Appendix A.

Band Number	*p* Value	Protein Name of the Top Hits	Fold Change ^a^	MW (kDa)
**46**	6.712440e-015	Cystatin	1.866	15.3
**36**	0.000441131	Salivary lipocalin & Ig lambdachain	1.558	27.8
**49**	0.000271945	Protein S100A12	0.602	12.4
**40**	4.243557e-007	Salivary lipocalin	0.348	24.1
**39**	4.994833e-011	Salivary lipocalin	0.324	24.8

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
