# Peer review of "Towards Understanding Non-Infectious Growth-Rate Retardation in Growing Pigs"

_proteomes, 2019, doi:10.3390/proteomes7030031_

Round 1

Reviewer 1 Report

The paper aimed to identify the pathways that could be modified in pigs with growth-rate retardation. For this, they used a gel-based proteomics approach on saliva samples, combined with mass spectrometry for the protein identification.

The Introduction section is well written and gives all the information to understand properly the context of this study.

The Material and Methods section is complete and provides enough details and/or reference for the readers. It should be interesting to explain why the use of both gel-based and label-free approaches (I am guessing to confirm or to obtain an identification when impossible using MALDI-TOF), and to justify the validation of S100A12 only.

Results

In the paragraph 3.4, it is mentioned that the PSM was used to determine the most abindant protein in the band. The authors should consider the use of emPAI that take into account the sequence lenght as a kind of normalisation.

With current label-free approaches, it is recommended to take these results with caution since the proteins determined as major is an assumption. In this case the last part about the validation using immunoblotting is of great importance.

In Material and Methods section, authors described the functional protein network using String software, but no mention of that is done in the results section. It should be interesting to determine if a biological process and/or a molecular function is up or down regulated in the study.

Discussion

Overall the Discussion section is well written and clear. It could be interesting to discuss the benefits of this approach, more mechanistic, regarding a more discriminant approach. Furthermore, the use of immunoblotting approach to validate should be discussed in regard with targeted MS methods such as PRM for instance.

Additional

The page numbers are not correct at the end of the document (after the table).

Reviewer 2 Report

The manuscript entitled "Towards understanding non-infectious growth-rate retardation in growing pigs" is overall well written, the experimental plan is clear and the results are well interpreted. There are, however, some points of concern:

-The real objective of the paper is not clear. From the title, the reader will expect a study aimed at explaining the proteomic changes related with the real causes of GRR in growing pigs. However, the main conclusion is that new biomarkers of GRR have been identified. I feel that the study design and the type of sample employed are more adapted for a "biomarker research" kind of paper, so I would suggest to change the title to something more in line with main conclusions, and maybe include in the introduction in what these kind of markers would be useful.

-In the study, it is not clear whether all animals tested were in the same farm. If this is the case, I would interpret that the proteome changes described in this study could be specific for the actual specific cause of GRR in that farm at that specific moment. In this case, results might be interpreted very carefully.

-The study would be much more robust if you could perform some measurements of any of the discovered biomarkers in additional samples (if only by WB?), other than the ones you have employed for the proteomic study, and preferably from another farm. If trends are confirmed, your conclusions will then be solid.

 - Animal groups are denominated "healthy" and "GRR" : I am not sure that GRR is a synomym of disease, and so I would change that to "control" vs "GRR". Plus, GRR is described as a "pathological condition" in the first sentence of the discussion. Again, I am not sure that GRR could be considered as a pathological condition by itself, maybe "pathophysiological" is more accurate??

-A band between approx. 30-45kDa is apparently more abundant in samples 1-5 in 1-DE gels. I wonder why this band was not excised and submitted for analysis.

-Blot normalization is not described in material and methods and the image of control loading protein/general blot stain is missing in figure 4a

-Were any of the parameters measured correlated? I wonder if there is any relationship between the markers you have measured that would help explain the results a bit further.

Round 2

Reviewer 2 Report

This reviewer is happy with the new version of the manuscript